# STOP MEMORIZING: A DATA-DEPENDENT REGULARIZATION FRAMEWORK FOR INTRINSIC PATTERN LEARNING

## ABSTRACT

Deep neural networks (DNNs) typically have enough capacity to fit random data by brute force even when conventional data-dependent regularizations focusing on the geometry of the features are imposed. We find out that the reason for this is the inconsistency between the enforced geometry and the standard softmax cross entropy loss. To resolve this, we propose a new framework for data-dependent DNN regularization, the Geometrically-Regularized-Self-Validating neural Networks (GRSVNet). During training, the geometry enforced on one batch of features is simultaneously validated on a separate batch using a validation loss consistent with the geometry. We study a particular case of GRSVNet, the Orthogonal-Low-rank Embedding (OLE)-GRSVNet, which is capable of producing highly discriminative features residing in orthogonal low-rank subspaces. Numerical experiments show that OLE-GRSVNet outperforms DNNs with conventional regularization when trained on real data. More importantly, unlike conventional DNNs, OLE-GRSVNet refuses to memorize random data or random labels, suggesting it only learns intrinsic patterns by reducing the memorizing capacity of the baseline DNN.

## 1 INTRODUCTION

It remains an open question why DNNs, typically with far more model parameters than training samples, can achieve such small generalization error. Previous work used various complexity measures from statistical learning theory, such as VC dimension (Vapnik, 1998), Radamacher complexity (Bartlett & Mendelson, 2002), and uniform stability (Bousquet & Elisseeff, 2002; Poggio et al., 2004), to provide an upper bound for the generalization error, suggesting that the effective capacity of DNNs, possibly with some regularization techniques, is usually limited.

However, the experiments by Zhang et al. (2017) showed that, even with data-independent regularization, DNNs can perfectly fit the training data when the true labels are replaced by random labels, or when the training data are replaced by Gaussian noise. This suggests that DNNs with data-independent regularization have enough capacity to "memorize" the training data. This poses an interesting question for network regularization design: is there a way for DNNs to refuse to (over)fit training samples with random labels, while exhibiting better generalization power than conventional DNNs when trained with true labels? Such networks are very important because they will extract only intrinsic patterns from the training data instead of memorizing miscellaneous details.

One would expect that data-dependent regularizations should be a better choice for reducing the memorizing capacity of DNNs. Such regularizations are typically enforced by penalizing the standard softmax cross entropy loss with an extra *geometric loss* which regularizes the feature geometry (Lezama et al., 2018; Zhu et al., 2018; Wen et al., 2016). However, regularizing DNNs with an extra geometric loss has two disadvantages: First, the output of the softmax layer, usually viewed as a probability distribution, is typically inconsistent with the feature geometry enforced by the geometric loss. Therefore, the geometric loss typically has a small weight to avoid jeopardizing the minimization of the softmax loss. Second, we find that DNNs with such regularization can still perfectly (over)fit random training samples or random labels. The reason is that the geometric loss (because of its small weight) is ignored and only the softmax loss is minimized.

This suggests that simply penalizing the softmax loss with a geometric loss is not sufficient to regularize DNNs. Instead, the softmax loss should be replaced by a *validation loss* that is consistent with the enforced geometry. More specifically, every training batch $B$ is split into two sub-batches,

the geometry batch $B^g$ and the validation batch $B^v$. The geometric loss $l_g$ is imposed on the features of $B^g$ for them to exhibit a desired geometric structure. A semi-supervised learning algorithm based on the proposed feature geometry is then used to generate a predicted label distribution for the validation batch, which combined with the true labels defines a validation loss on $B^v$. The total loss on the training batch $B$ is then defined as the weighted sum $l = l_g + \lambda l_v$. Because the predicted label distribution on $B^v$ is based on the enforced geometry, the geometric loss $l_g$ can no longer be neglected. Therefore, $l_g$ and $l_v$ will be minimized simultaneously, i.e., the geometry is correctly enforced (small $l_g$) and it can be used to predict validation samples (small $l_v$). We call such DNNs Geometrically-Regularized-Self-Validating neural Networks (GRSVNets). See Figure 1a for a visual illustration of the network architecture.

GRSVNet is a general architecture because every consistent geometry/validation pair can fit into this framework as long as the loss functions are differentiable. In this paper, we focus on a particular type of GRSVNet, the Orthogonal-Low-rank-Embedding-GRSVNet (OLE-GRSVNet). More specifically, we impose the OLE loss (Qiu & Sapiro, 2015) on the geometry batch to produce features residing in orthogonal subspaces, and we use the principal angles between the validation features and those subspaces to define a predicted label distribution on the validation batch. We prove that the loss function obtains its minimum if and only if the subspaces of different classes spanned by the features in the geometry batch are orthogonal, and the features in the validation batch reside perfectly in the subspaces corresponding to their labels (see Figure 1e). We show in our experiments that OLE-GRSVNet has better generalization performance when trained on real data, but it refuses to memorize the training samples when given random training data or random labels, which suggests that OLE-GRSVNet effectively learns intrinsic patterns.

Our contributions can be summarized as follows:

- We proposed a general framework, GRSVNet, to effectively impose data-dependent DNN regularization. The core idea is the self-validation of the enforced geometry with a consistent validation loss on a separate batch of features.

- We study a particular case of GRSVNet, OLE-GRSVNet, that can produce highly discriminative features: samples from the same class belong to a low-rank subspace, and the subspaces for different classes are orthogonal.

- OLE-GRSVNet achieves better generalization performance when compared to DNNs with conventional regularizers. And more importantly, unlike conventional DNNs, OLE-GRSVNet refuses to fit the training data (i.e., with a training error close to random guess) when the training data or the training labels are randomly generated. This implies that OLE-GRSVNet never memorizes the training samples, only learns intrinsic patterns.

## 2 RELATED WORK

Many data-dependent regularizations focusing on feature geometry have been proposed for deep learning (Lezama et al., 2018; Zhu et al., 2018; Wen et al., 2016). The center loss (Wen et al., 2016) produces compact clusters by minimizing the Euclidean distance between features and their class centers. LDMNet (Zhu et al., 2018) extracts features sampling a collection of low dimensional manifolds. The OLE loss (Lezama et al., 2018; Qiu & Sapiro, 2015) increases inter-class separation and intra-class similarity by embedding inputs into orthogonal low-rank subspaces. However, as mentioned in Section 1, these regularizations are imposed by adding the geometric loss to the softmax loss, which, when viewed as a probability distribution, is typically not consistent with the desired geometry. Our proposed GRSVNet instead uses a validation loss based on the regularized geometry so that the predicted label distribution has a meaningful geometric interpretation.

The way in which GRSVNets impose geometric loss and validation loss on two separate batches of features extracted with two identical baseline DNNs bears a certain resemblance to the siamese network architecture (Chopra et al., 2005) used extensively in metric learning (Cheng et al., 2016; Hadsell et al., 2006; Hu et al., 2014; Schroff et al., 2015; Sun et al., 2014). The difference is, unlike contrastive loss (Hadsell et al., 2006) and triplet loss (Schroff et al., 2015) in metric learning, the feature geometry is explicitly regularized in GRSVNets, and a representation of the geometry, e.g., basis of the low-rank subspace, can be later used directly for the classification of test data.

Our work is also related to two recent papers (Zhang et al., 2017; Arpit et al., 2017) addressing the memorization of DNNs. Zhang et al. (2017) empirically showed that conventional DNNs, even with

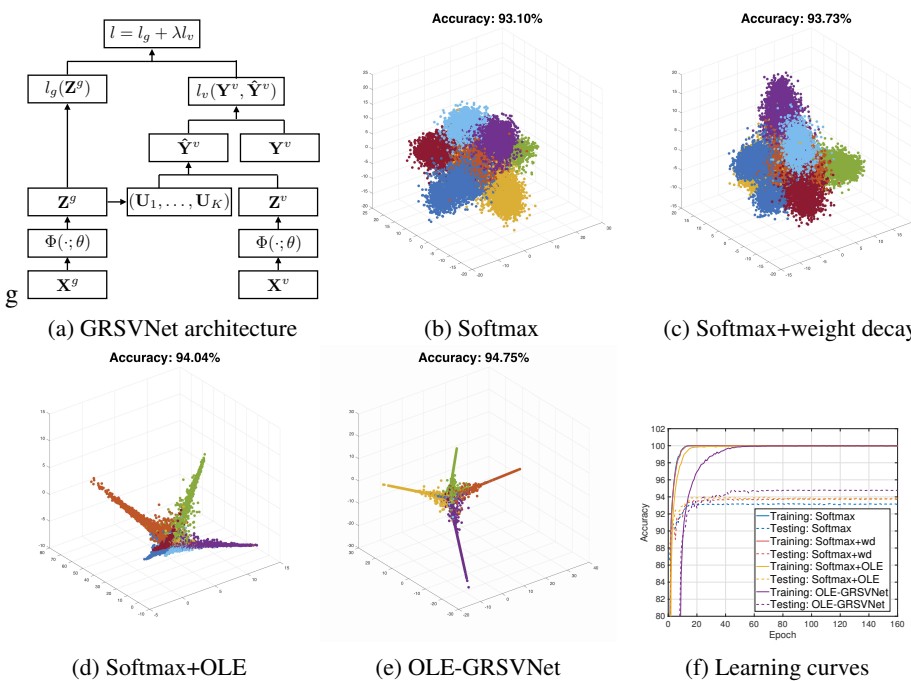

Figure 1: GRSVNet architecture and the results of different networks with the same VGG-11 baseline achitecture on the SVHN dataset with real data and real labels. (a) GRSVNet architecture (better understood in its special case OLE-GRSVNet detailed in Section 3). (b)-(e) Features of the test data learned by different networks visualized in 3D using PCA. Note that for OLE-GRSVNet, only four classes (out of ten) have nonzero 3D embedding (Theorem 2). (f) Training/testing accuracy.

data-independent regularization, are fully capable of memorizing random labels or random data. Arpit et al. (2017) argued that DNNs trained with stochastic gradient descent (SGD) tend to fit patterns first before memorizing miscellaneous details, suggesting that memorization of DNNs depends also on the data itself, and SGD with early stopping is a valid strategy in conventional DNN training. We demonstrate in our paper that when data-dependent regularization is imposed in accordance with the validation, GRSVNets will **never** memorize random labels or random data, and only extracts intrinsic patterns. An explanation of this phenomenon is provided in Section 4.

## 3 GRSVNET AND ITS SPECIAL CASE: OLE-GRSVNET

As pointed out in Section 1, the core idea of GRSVNet is to self-validate the geometry using a consistent validation loss. To contextualize this idea, we study a particular case, OLE-GRSVNet, where the regularized feature geometry is orthogonal low-rank subspaces, and the validation loss is defined by the principal angles between the validation features and the subspaces.

### 3.1 OLE LOSS

The OLE loss was originally proposed by Qiu & Sapiro (2015). Consider a $K$-way classification problem. Let $\mathbf{X} = [\boldsymbol{x}_1, \ldots, \boldsymbol{x}_N] \in \mathbb{R}^{d \times N}$ be a collection of data points $\{\boldsymbol{x}_i\}_{i=1}^N \subset \mathbb{R}^d$. Let $\mathbf{X}_c$ denote the submatrix of $\mathbf{X}$ formed by inputs of the $c$-th class. Qiu & Sapiro (2015) proposed to learn a linear transformation $\mathbf{T} : \mathbb{R}^d \to \mathbb{R}^d$ that maps data from the same class $\mathbf{X}_c$ into a low-rank subspace, while mapping the entire data $\mathbf{X}$ into a high-rank linear space. This is achieved by solving:

$$\min_{\mathbf{T}:\mathbb{R}^d \to \mathbb{R}^d} \sum_{c=1}^{K} \|\mathbf{T}\mathbf{X}_c\|_* - \|\mathbf{T}\mathbf{X}\|_*, \quad \text{s.t.} \ \|\mathbf{T}\|_2 = 1, \tag{1}$$

where $\| \cdot \|_*$ is the matrix nuclear norm, which is a convex lower bound of the rank function on the unit ball in the operator norm (Recht et al., 2010). The norm constraint $\|\mathbf{T}\|_2 = 1$ is imposed to

avoid the trivial solution $\mathbf{T} = \mathbf{0}$. It is proved by Qiu & Sapiro (2015) that the OLE loss (1) is always nonnegative, and the global optimum value 0 is obtained if $\mathbf{TX}_c \perp \mathbf{TX}_{c'}, \forall c \neq c'$.

Lezama et al. (2018) later used OLE loss as a data-dependent regularization for deep learning. Given a baseline DNN that maps a batch of inputs $\mathbf{X}$ into the features $\mathbf{Z} = \Phi(\mathbf{X}; \theta)$, the OLE loss on $\mathbf{Z}$ is

$$l_g(\mathbf{Z}) = \sum_{c=1}^{K} \|\mathbf{Z}_c\|_* - \|\mathbf{Z}\|_* = \sum_{c=1}^{K} \|\Phi(\mathbf{X}_c; \theta)\|_* - \|\Phi(\mathbf{X}; \theta)\|_*. \tag{2}$$

The OLE loss is later combined with the standard softmax loss for training, and we will henceforth call such network "softmax+OLE." Softmax+OLE significantly improves the generalization performance, but it suffers from two problems because of the inconsistency between the softmax loss and the OLE loss: First, the learned features no longer exhibit the desired geometry of orthogonal low-rank subspaces. Second, as will be shown in Section 4, softmax+OLE is still capable of memorizing random data or random labels, i.e., it has not reduced the memorizing capacity of DNNs.

### 3.2 OLE-GRSVNET

We will now explain how to incorporate OLE loss into the GRSVNet framework. First, let us better understand the geometry enforced by the OLE loss by stating the following theorem.

**Theorem 1.** *Let* $\mathbf{Z} = [\mathbf{Z}_1, \ldots, \mathbf{Z}_c]$ *be a horizontal concatenation of matrices* $\{\mathbf{Z}_c\}_{c=1}^{K}$. *The OLE loss* $l_g(\mathbf{Z})$ *defined in* (2) *is always nonnegative. Moreover,* $l_g(\mathbf{Z}) = 0$ *if and only if* $\mathbf{Z}_c^* \mathbf{Z}_{c'} = \mathbf{0}, \forall c \neq c'$, *i.e., the column spaces of* $\mathbf{Z}_c$ *and* $\mathbf{Z}_{c'}$ *are orthogonal.*

The proof of Theorem 1, as well as those of the remaining theorems, is detailed in the Appendix. Note that Theorem 1, which ensures that the OLE loss is minimized if and **only** if features of different classes are orthogonal, is a much stronger result than that by Qiu & Sapiro (2015). We then need to define a validation loss $l_v$ that is consistent with the geometry enforced by $l_g$. A natural choice would be the principal angles between the validation features and the subspaces spanned by $\{\mathbf{Z}_c\}_{c=1}^{K}$.

Now we detail the architecture for OLE-GRSVNet. Given a baseline DNN, we split every training batch $\mathbf{X} \in \mathbb{R}^{d \times |B|}$ into two sub-batches, the geometry batch $\mathbf{X}^g \in \mathbb{R}^{d \times |B_g|}$ and the validation batch $\mathbf{X}^v \in \mathbb{R}^{d \times |B_v|}$, which are mapped by the same baseline DNN into features $\mathbf{Z}^g = \Phi(\mathbf{X}^g; \theta)$ and $\mathbf{Z}^v = \Phi(\mathbf{X}^v; \theta)$. The OLE loss $l_g(\mathbf{Z}^g)$ is imposed on the geometry batch to ensure $\text{span}(\mathbf{Z}_c^g)$ are orthogonal low-rank subspaces, where $\text{span}(\mathbf{Z}_c^g)$ is the column space of $\mathbf{Z}_c^g$. Let $\mathbf{Z}_c^g = \mathbf{U}_c \mathbf{\Sigma}_c \mathbf{V}_c^*$ be the (compact) singular value decomposition (SVD) of $\mathbf{Z}_c^g$, then the columns of $\mathbf{U}_c$ form an orthonormal basis of $\text{span}(\mathbf{Z}_c^g)$. For any feature $\boldsymbol{z} = \Phi(\boldsymbol{x}; \theta) \in \mathbf{Z}^v$ in the validation batch, its projection onto the subspace $\text{span}(\mathbf{Z}_c^g)$ is $\text{proj}_c(\boldsymbol{z}) = \mathbf{U}_c \mathbf{U}_c^* \boldsymbol{z}$. The cosine similarity between $\boldsymbol{z}$ and $\text{proj}_c(\boldsymbol{z})$ is then defined as the (unnormalized) probability of $\boldsymbol{x}$ belonging to class $c$, i.e.,

$$\hat{y}_c = \mathbf{P}(\boldsymbol{x} \in c) \triangleq \left\langle \boldsymbol{z}, \frac{\text{proj}_c(\boldsymbol{z})}{\max\left(\|\text{proj}_c(\boldsymbol{z})\|, \varepsilon\right)} \right\rangle \bigg/ \sum_{c'=1}^{K} \left\langle \boldsymbol{z}, \frac{\text{proj}_{c'}(\boldsymbol{z})}{\max\left(\|\text{proj}_{c'}(\boldsymbol{z})\|, \varepsilon\right)} \right\rangle, \tag{3}$$

where a small $\varepsilon$ is chosen for numerical stability. The validation loss for $\boldsymbol{x}$ is then defined as the cross entropy between the predicted distribution $\hat{\boldsymbol{y}} = (\hat{y}_1, \ldots, \hat{y}_K)^T \in \mathbb{R}^K$ and the true label $y \in \{1, \ldots, K\}$. More specifically, let $\mathbf{Y}^v \in \mathbb{R}^{1 \times |B_v|}$ and $\hat{\mathbf{Y}}^v \in \mathbb{R}^{K \times |B_v|}$ be the collection of true labels and predicted label distributions on the validation batch, then the validation loss is defined as

$$l_v(\mathbf{Y}^v, \hat{\mathbf{Y}}^v) = \frac{1}{|B_v|} \sum_{\boldsymbol{x} \in \mathbf{X}^v} H(\delta_y, \hat{\boldsymbol{y}}) = -\frac{1}{|B_v|} \sum_{\boldsymbol{x} \in \mathbf{X}^v} \log \hat{y}_y, \tag{4}$$

where $\delta_y$ is the Dirac distribution at label $y$, and $H(\cdot, \cdot)$ is the cross entropy between two distributions. The empirical loss $l$ on the training batch $\mathbf{X}$ is then defined as

$$l(\mathbf{X}, \mathbf{Y}) = l([\mathbf{X}^g, \mathbf{X}^v], [\mathbf{Y}^g, \mathbf{Y}^v]) = l_g(\mathbf{Z}^g) + \lambda l_v(\mathbf{Y}^v, \hat{\mathbf{Y}}^v). \tag{5}$$

See Figure 1a for a visual illustration of the OLE-GRSVNet architecture. Because of the consistency between $l_g$ and $l_v$, we have the following theorem:

**Theorem 2.** *For any* $\lambda > 0$, *and any geometry/validation splitting of* $\mathbf{X} = [\mathbf{X}^g, \mathbf{X}^v]$ *satisfying* $\mathbf{X}^v$ *contains at least one sample for each class, the empirical loss function defined in* (5) *is always nonnegative.* $l(\mathbf{X}, \mathbf{Y}) = 0$ *if and only if both of the following conditions hold true:*

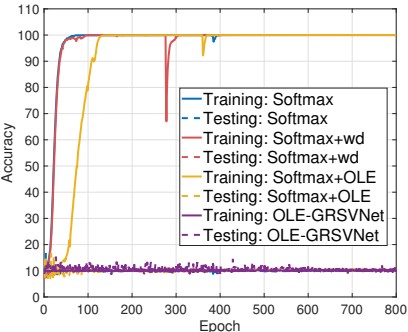 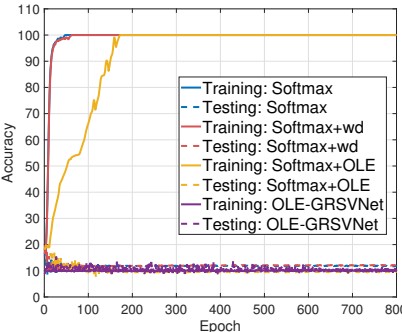

(a) Training/testing accuracy with random labels    (b) Training/testing accuracy with random data

Figure 2: Training and testing accuracy of different networks on the SVHN dataset with random labels or random data (Gaussian noise). Note that softmax, sotmax+wd, and softmax+OLE can all perfectly (over)fit the random training data or training data with random labels. However, OLE-GRSVNet refuses to fit the training data when there is no intrinsically learnable patterns.

- *The features of the geometry batch belonging to different classes are orthogonal, i.e.,* $\mathrm{span}(\mathbf{Z}_c^g) \perp \mathrm{span}(\mathbf{Z}_{c'}^g), \forall c \neq c'$

- *For every datum* $\boldsymbol{x} \in \mathbf{X}_c^v$, *i.e.,* $\boldsymbol{x}$ *belongs to class $c$ in the validation batch, its feature* $\boldsymbol{z} = \Phi(\boldsymbol{x}; \theta)$ *belongs to* $\mathrm{span}(\mathbf{Z}_c^g)$.

*Moreover, if $l < \infty$, then* $\mathrm{rank}(\mathrm{span}(\mathbf{Z}_c^g)) \geq 1, \forall c$, *i.e.,* $\Phi(\cdot; \theta)$ *does not trivially map data into $\mathbf{0}$.*

**Remark**: The requirement that $\lambda > 0$ is crucial in Theorem 2, because otherwise the network can map every input into $\mathbf{0}$ and achieve the minimum. This is validated in our numerical experiments.

After the training process has finished, we can then map the entire training data $\mathbf{X}^{\mathrm{all}} = [\mathbf{X}_1^{\mathrm{all}}, \ldots, \mathbf{X}_K^{\mathrm{all}}]$ (or a random portion of $\mathbf{X}^{\mathrm{all}}$) into their features $\mathbf{Z}^{\mathrm{all}} = \Phi(\mathbf{X}^{\mathrm{all}}; \theta^*)$, where $\theta^*$ is the learned parameter. The low-rank subspace $\mathrm{span}(\mathbf{Z}_c^{\mathrm{all}})$ for class $c$ can be obtained via the SVD of $\mathbf{Z}_c^{\mathrm{all}}$. The label of a test datum $\boldsymbol{x}$ is then determined by the principal angles between $\boldsymbol{z} = \Phi(\boldsymbol{x}; \theta^*)$ and $\{\mathrm{span}(\mathbf{Z}_c^{\mathrm{all}})\}_{c=1}^K$.

## 4  TWO TOY EXPERIMENTS

Before delving into the implementation details of OLE-GRSVNet, we first present two toy experiments to illustrate our proposed framework. We use VGG-11 (Simonyan & Zisserman, 2014) as the baseline architecture, and compare the performance of the following four DNNs: (a) The baseline network with a softmax classifier (softmax). (b) VGG-11 with weight decay (softmax+wd). (c) VGG-11 regularized by penalizing the softmax loss with the OLE loss (softmax+OLE) (d) OLE-GRSVNet.

We first train these four DNNs on the Street View House Numbers (SVHN) dataset with the original data and labels without data augmentation. The test accuracy and the PCA embedding of the learned test features are shown in Figure 1. OLE-GRSVNet has the highest test accuracy among the comparing DNNs. Moreover, because of the consistency between the geometric loss and the validation loss, the test features produced by OLE-GRSVNet are even more discriminative than softmax+OLE: features of the same class reside in a low-rank subspace, and different subspaces are (almost) orthogonal. Note that in Figure 1e, features of only four classes out of ten (though ideally it should be three) have nonzero 3D embedding (Theorem 2).

Next, we train the same networks, without changing hyperparameters, on the SVHN dataset with either (a) randomly generated labels, or (b) random training data (Gaussian noise). We train the DNNs for 800 epochs to ensure their convergence, and the learning curves of training/testing accuracy are shown in Figure 2. Note that the baseline DNN, with either data-independent or conventional data-dependent regularization, can perfectly (over)fit the training data, while OLE-GRSVNet refuses to memorize the training data when there are no intrinsically learnable patterns.

In another experiment, we generate three classes of one-dimensional data in $\mathbb{R}^{10}$: the data points in the $i$-th class are i.i.d. samples from the Gaussian distribution with the standard deviation in

the $i$-th coordinate 50 times larger than other coordinates. Each class has 500 data points, and we randomly shuffle the class labels after generation. We then train a multilayer perceptron (MLP) with 128 neurons in each layer for 2000 epochs to classify these low dimensional data with random labels. We found out that only three layers are needed to perfectly classify these data when using a softmax classifier. However, after incrementally adding more layers to the baseline MLP, we found out that OLE-GRSVNet still refuses to memorize the random labels even for 100-layer MLP. This further suggests that OLE-GRSVNet refuses to memorize training data by brute force when there is no intrinsic patterns in the data. A visual illustration of this experiment is shown in the Appendix.

We provide an intuitive explanation for why OLE-GRSVNet can generalize very well when given true labeled data but refuses to memorize random data or random labels. By Theorem 2, we know that OLE-GRSVNet obtains its global minimum if and only if the features of every random training batch exhibit the same orthogonal low-rank-subspace structure. This essentially implies that OLE-GRSVNet is implicitly conducting $O(N^{|B|})$-fold data augmentation, where $N$ is the number of training data, and $|B|$ is the batch size, while conventional data augmentation by the manipulation of the inputs, e.g., random cropping, flipping, etc., is typically $O(N)$. This poses a very interesting question: Does it mean that OLE-GRSVNet can also memorize random data if the baseline DNN has exponentially many model parameters? Or is it because of the learning algorithm (SGD) that prevents OLE-GRSVNet from learning a decision boundary too complicated for classifying random data? Answering this question will be the focus of our future research.

## 5 IMPLEMENTATION DETAILS OF OLE-GRSVNET

Most of the operations in the computational graph of OLE-GRSVNet (Figure 1a) explained in Section 3 are basic matrix operations. The only two exceptions are the OLE loss ($\mathbf{Z}_g \to l^g((\mathbf{Z}^g))$) and the SVD ($\mathbf{Z}^g \to (\mathbf{U}_1, \dots, \mathbf{U}_K)$). We hereby specify their forward and backward propagations.

### 5.1 BACKWARD PROPAGATION OF THE OLE LOSS

According to the definition of the OLE loss in (2), we only need to find a (sub)gradient of the nuclear norm to back-propagate the OLE loss. The characterization of the subdifferential of the nuclear norm is explained by Watson (1992). More specifically, assuming $m \geq n$ for simplicity, let $\mathbf{U} \in \mathbb{R}^{m \times m}$, $\mathbf{\Sigma} \in \mathbb{R}^{m \times n}$, $\mathbf{V} \in \mathbb{R}^{n \times n}$ be the SVD of a rank-$s$ matrix $\mathbf{A}$. Let $\mathbf{U} = [\mathbf{U}^{(1)}, \mathbf{U}^{(2)}]$, $\mathbf{V} = [\mathbf{V}^{(1)}, \mathbf{V}^{(2)}]$ be the partition of $\mathbf{U}$, $\mathbf{V}$, respectively, where $\mathbf{U}^{(1)} \in \mathbb{R}^{m \times s}$ and $\mathbf{V}^{(1)} \in \mathbb{R}^{n \times s}$, then the subdifferential of the nuclear norm at $\mathbf{A}$ is:

$$\partial \|\mathbf{A}\|_* = \left\{ \mathbf{U}^{(1)}\mathbf{V}^{(1)*} + \mathbf{U}^{(2)}\mathbf{W}\mathbf{V}^{(2)*}, \quad \forall \mathbf{W} \in \mathbb{R}^{(m-s) \times (n-s)} \text{ with } \|\mathbf{W}\|_2 \leq 1 \right\}, \quad (6)$$

where $\| \cdot \|_2$ is the spectral norm. Note that to use (6), one needs to identify the rank-$s$ column space of $\mathbf{A}$, i.e., $\mathrm{span}(\mathbf{U}^{(1)})$ to find a subgradient, which is not necessarily easy because of the existence of numerical error. Lezama et al. (2018) intuitively truncated the numerical SVD with a small parameter chosen a priori to ensure the numerical stability. We show in the following theorem using the backward stability of SVD that such concern is, in theory, not necessary.

**Theorem 3.** *Let $\mathbf{U}^\varepsilon, \mathbf{\Sigma}^\varepsilon, \mathbf{V}^\epsilon$ be the numerically computed reduced SVD of $\mathbf{A} \in \mathbb{R}^{m \times n}$, i.e., $\mathbf{U}^\varepsilon \in \mathbb{R}^{m \times n}$, $\mathbf{V}^\varepsilon \in \mathbb{R}^{n \times n}$, $(\mathbf{U}^\varepsilon + \delta\mathbf{U}^\varepsilon)\mathbf{\Sigma}^\varepsilon(\mathbf{V}^\varepsilon + \delta\mathbf{V}^\varepsilon)^* = \mathbf{A} + \delta\mathbf{A} = \mathbf{A}^\varepsilon$, and $\|\delta\mathbf{U}\|_2, \|\delta\mathbf{V}\|_2, \|\delta\mathbf{A}\|_2$ are all $O(\varepsilon)$, where $\varepsilon$ is the machine error. If $\mathrm{rank}(\mathbf{A}) = s \leq n$, and the smallest singular value $\sigma_s(\mathbf{A})$ of $\mathbf{A}$ satisfies $\sigma_s(\mathbf{A}) \geq \eta > 0$, we have*

$$\mathrm{d}(\mathbf{U}^\varepsilon\mathbf{V}^{\varepsilon*}, \partial\|\mathbf{A}\|_*) = O(\varepsilon/\eta). \quad (7)$$

However, in practice we did observe that using a small threshold ($10^{-6}$ in this work) to truncate the numerical SVD can speed up the convergence, especially in the first few epochs of training. With the help of Theorem 3, we can easily find a stable subgradient of the OLE loss in (2).

### 5.2 FORWARD AND BACKWARD PROPAGATION OF $\mathbf{Z}^g \to (\mathbf{U}_1, \dots, \mathbf{U}_K)$

Unlike the computation of the subgradient in Theorem 3, we have to threshold the singular vectors of $\mathbf{Z}_c^g$, because the desired output $\mathbf{U}_c$ should be an orthonormal basis of the low-rank subspace $\mathrm{span}(\mathbf{Z}_c^g)$. In the forward propagation, we threshold the singular vectors $\mathbf{U}_c$ such that the smallest singular value is at least $1/10$ of the largest singular value.

As for the backward propagation, one needs to know the Jacobian of SVD, which has been explained by Papadopoulo & Lourakis (2000). Typically, for a matrix $\mathbf{A} \in \mathbb{R}^{n \times n}$, computing the Jacobian of the SVD of $\mathbf{A}$ involves solving a total of $O(n^4)$ $2 \times 2$ linear systems. We have not implemented the backward propagation of SVD in this work because this involves technical implementation with CUDA API. In our current implementation, the node $(\mathbf{U}_1, \ldots, \mathbf{U}_K)$ is detached from the computational graph during back propagation, i.e., the validation loss $l_v$ is only propagated back through the path $l_v \rightarrow \hat{\mathbf{Y}}^v \rightarrow \mathbf{Z}^v \rightarrow \theta$. Our rational is this: The validation loss $l_v$ can be propagated back through two paths: $l_v \rightarrow \hat{\mathbf{Y}}^v \rightarrow \mathbf{Z}^v \rightarrow \theta$ and $l_v \rightarrow \hat{\mathbf{Y}}^v \rightarrow (\mathbf{U_1}, \ldots, \mathbf{U}_K) \rightarrow \mathbf{Z}^g \rightarrow \theta$. The first path will modify $\theta$ so that $\mathbf{Z}_c^v$ moves closer to $\mathbf{U}_c$, while the second path will move $\mathbf{U}_c$ closer to $\mathbf{Z}_c^v$. Cutting off the second path when computing the gradient might decrease the speed of convergence, but numerical experiments suggest that the training process is still well-behaved under such simplification. With such simplification, the only extra computation is the SVD of a mini-batch of features, which is negligible (<5%) when compared to the time of training the baseline network.

## 6 EXPERIMENTAL RESULTS

In this section, we demonstrate the superiority of OLE-GRSVNet when compared to conventional DNNs in two aspects: (a) It has greater generalization power when trained on true data and true labels. (b) Unlike conventionally regularized DNNs, OLE-GRSVNet refuses to memorize the training samples when given random training data or random labels.

We use similar experimental setup as in Section 4. The same four modifications to three baseline architectures (VGG-11,16,19 (Simonyan & Zisserman, 2014)) are considered: (a) **Softmax**. (b) **Softmax+wd**. (c) **Softmax+OLE** (d) **OLE-GRSVNet**. The performance of the networks are tested on the following datasets:

- **MNIST**. The MNIST dataset contains $28 \times 28$ grayscale images of digits from 0 to 9. There are 60,000 training samples and 10,000 testing samples. No data augmentation was used.

- **SVHN**. The Street View House Numbers (SVHN) dataset contains $32 \times 32$ RGB images of digits from 0 to 9. The training and testing set contain 73,257 and 26,032 images respectively. No data augmentation was used.

- **CIFAR**. This dataset contains $32 \times 32$ RGB images of ten classes, with 50,000 images for training and 10,000 images for testing. We use "**CIFAR+**" to denote experiments on CIFAR with data augmentation: 4 pixel padding, $32 \times 32$ random cropping and horizontal flipping.

### 6.1 TRAINING DETAILS

All networks are trained from scratch with the "Xavier" initialization (Glorot & Bengio, 2010). SGD with Nesterov momentum 0.9 is used for the optimization, and the batch size is set to 200 (a 100/100 split for geometry/validation batch is used in OLE-GRSVNet). We set the initial learning rate to 0.01, and decrease it ten-fold at 50% and 75% of the total training epochs. For the experiments with true labels, all networks are trained for 100, 160 epochs for MNIST, SVHN, respectively. For CIFAR, we train the networks for 200, 300, 400 epochs for VGG-11, VGG16, VGG-19, respectively. In order to ensure the convergence of SGD, all networks are trained for 800 epochs for the experiments with random labels. The mean accuracy after five independent trials is reported.

The weight decay parameter is always set to $\mu = 10^{-4}$. The weight for the OLE loss in "softmax+OLE" is chosen according to Lezama et al. (2018). More specifically, it is set to 0.5 for MNIST and SVHN, 0.5 for CIFAR with VGG-11 and VGG-16, and 0.25 for CIFAR with VGG-19. For OLE-GRSVNet, the parameter $\lambda$ in (5) is determined by cross-validation. More specifically, we set $\lambda = 10$ for MNIST, $\lambda = 5$ for SVHN and CIFAR with VGG-11 and VGG-16, and $\lambda = 1$ for CIFAR with VGG-19.

### 6.2 TESTING/TRAINING PERFORMANCE WHEN TRAINED ON DATA WITH REAL OR RANDOM LABELS

Table 1 reports the performance of the networks trained on the original data with real or randomly generated labels. The numbers without parentheses are the percentage accuracies on the **test** data when networks are trained with **real** labels, and the numbers enclosed in parentheses are the accuracies on the **training** data when given **random** labels. Accuracies on the training data with real labels

| Dataset | VGG | Testing (training) accuracy (%) | | | |
|---------|-----|---------|-----------|------------|-----------|
| | | Softmax | Softmax+wd | Softmax+OLE | OLE-GRSVNet |
| MNIST | 11 | 99.40 (100.00) | 99.47 (100.00) | 99.49 (100.00) | **99.57 (9.93)** |
| SVHN | 11 | 93.10 (99.99) | 93.73 (100.00) | 94.04 (99.99) | **94.75 (9.75)** |
| CIFAR | 11 | 81.81 (100.00) | 81.87 (100.00) | 82.04 (99.95) | **85.29 (9.97)** |
| CIFAR | 16 | 83.37 (100.00) | 83.97 (99.99) | 84.35 (99.96) | **87.44 (10.13)** |
| CIFAR | 19 | 83.56 (99.99) | 84.21 (99.97) | 84.71 (99.96) | **86.69 (9.86)** |
| CIFAR+ | 11 | 89.52 (99.98) | 89.68 (99.98) | 90.04 (99.93) | **90.58 (10.05)** |
| CIFAR+ | 16 | 91.21 (99.96) | 91.29 (99.96) | 91.40 (99.92) | **92.15 (9.94)** |
| CIFAR+ | 19 | 91.19 (99.96) | 91.53 (99.95) | **91.67** (99.91) | 91.65 **(10.07)** |

Table 1: Testing or training accuracies when trained on training data with real or random labels. The numbers without parentheses are the percentage accuracies on the **testing** data when networks are trained with **real** labels. The numbers enclosed in parentheses are the accuracies on the **training** data when networks are trained with **random** labels. The mean accuracy after five independent trials is reported. This suggests that OLE-GRSVNet outperforms conventional DNNs on the testing data when trained with real labels. Moreover, unlike conventional DNNs, OLE-GRSVNet refuses to memorize the training data when trained with random labels.

(always 100%) and accuracies on the test data with random labels (always close to 10%) are omitted from the table. As we can see, similar to the experiment in Section 4, when trained with real labels, OLE-GRSVNet exhibits better generalization performance than the competing networks. But when trained with random labels, OLE-GRSVNet refuses to memorize the training samples like the other networks because there are no intrinsically learnable patterns. This is still the case even if we increase the number of training epochs to 2000.

We point out that by combining different regularization and tuning the hyperparameters, the test error of conventional DNNs can indeed be reduced. For example, if we combine weight decay, conventional OLE regularization, batch normalization, data augmentation, and increase the learning rate from $0.01$ to $0.1$, the test accuracy of CIFAR can be pushed to $91.02\%$. However, this does not change the fact that such network can still perfectly memorize training samples when given random labels. This corroborates the claim by Zhang et al. (2017) that conventional regularization appears to be more of a tuning parameter instead of playing an essential role in reducing network capacity.

## 7 CONCLUSION AND FUTURE WORK

We proposed a general framework, GRSVNet, for data-dependent DNN regularization. The core idea is the self-validation of the enforced geometry on a separate batch using a validation loss consistent with the geometric loss, so that the predicted label distribution has a meaningful geometric interpretation. In particular, we study a special case of GRSVNet, OLE-GRSVNet, which is capable of producing highly discriminative features: samples from the same class belong to a low-rank subspace, and the subspaces for different classes are orthogonal. When trained on benchmark datasets with real labels, OLE-GRSVNet achieves better test accuracy when compared to DNNs with different regularizations sharing the same baseline architecture. More importantly, unlike conventional DNNs, OLE-GRSVNet refuses to memorize and overfit the training data when trained on random labels or random data. This suggests that OLE-GRSVNet effectively reduces the memorizing capacity of DNNs, and it only extracts intrinsically learnable patterns from the data.

Although we provided some intuitive explanation as to why GRSVNet generalizes well on real data and refuses overfitting random data, there are still open questions to be answered. For example, what is the minimum representational capacity of the baseline DNN (i.e., number of layers and number of units) to make even GRSVNet trainable on random data? Or is it because of the learning algorithm (SGD) that prevents GRSVNet from learning a decision boundary that is too complicated for random samples? Moreover, we still have not answered why conventional DNNs, while fully capable of memorizing random data by brute force, typically find generalizable solutions on real data. These questions will be the focus of our future work.

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

## APPENDIX A    PROOF OF THEOREM 1

It suffices to prove the case when $K = 2$, as the case for larger $K$ can be proved by induction. In order to simplify the notation, we restate the original theorem for $K = 2$:

**Theorem.** *Let $\mathbf{A} \in \mathbb{R}^{N \times m}$ and $\mathbf{B} \in \mathbb{R}^{N \times n}$ be matrices of the same row dimensions, and $[\mathbf{A}, \mathbf{B}] \in \mathbb{R}^{N \times (m+n)}$ be the concatenation of $\mathbf{A}$ and $\mathbf{B}$. We have*

$$\|[\mathbf{A}, \mathbf{B}]\|_* \le \|\mathbf{A}\|_* + \|\mathbf{B}\|_*. \tag{8}$$

*Moreover, the equality holds if and only if $\mathbf{A}^* \mathbf{B} = \mathbf{0}$, i.e., the column spaces of $\mathbf{A}$ and $\mathbf{B}$ are orthogonal.*

*Proof.* The inequality (8) and the sufficient condition for the equality to hold is easy to prove. More specifically,

$$\|[\mathbf{A}, \mathbf{B}]\|_* = \|[\mathbf{A}, \mathbf{0}] + [\mathbf{0}, \mathbf{B}]\|_* \le \|[\mathbf{A}, \mathbf{0}]\|_* + \|[\mathbf{0}, \mathbf{B}]\|_* = \|\mathbf{A}\|_* + \|\mathbf{B}\|_*. \tag{9}$$

Moreover, if $\mathbf{A}^* \mathbf{B} = \mathbf{0}$, then

$$|[\mathbf{A}, \mathbf{B}]|^2 = [\mathbf{A} \quad \mathbf{B}]^* [\mathbf{A} \quad \mathbf{B}] = \begin{bmatrix} \mathbf{A}^* \mathbf{A} & \mathbf{A}^* \mathbf{B} \\ \mathbf{B}^* \mathbf{A} & \mathbf{B}^* \mathbf{B} \end{bmatrix} = \begin{bmatrix} \mathbf{A}^* \mathbf{A} & \mathbf{0} \\ \mathbf{0} & \mathbf{B}^* \mathbf{B} \end{bmatrix} = \begin{bmatrix} |\mathbf{A}| & \mathbf{0} \\ \mathbf{0} & |\mathbf{B}| \end{bmatrix}^2, \tag{10}$$

where $|\mathbf{A}| = (\mathbf{A}^* \mathbf{A})^{\frac{1}{2}}$. Therefore,

$$\|[\mathbf{A}, \mathbf{B}]\|_* = \text{Tr}\left(|[\mathbf{A}, \mathbf{B}]|\right) = \text{Tr}\left(\begin{bmatrix} |\mathbf{A}| & \mathbf{0} \\ \mathbf{0} & |\mathbf{B}| \end{bmatrix}\right) = \text{Tr}(|\mathbf{A}|) + \text{Tr}(|\mathbf{B}|) = \|\mathbf{A}\|_* + \|\mathbf{B}\|_*. \tag{11}$$

Next, we show the necessary condition for the equality to hold, i.e.,

$$\|[\mathbf{A}, \mathbf{B}]\|_* = \|\mathbf{A}\|_* + \|\mathbf{B}\|_* \implies \mathbf{A}^* \mathbf{B} = \mathbf{0}. \tag{12}$$

Let $\begin{bmatrix} \mathbf{E} & \mathbf{G} \\ \mathbf{G}^* & \mathbf{F} \end{bmatrix} = \begin{bmatrix} \mathbf{A}^* \mathbf{A} & \mathbf{A}^* \mathbf{B} \\ \mathbf{B}^* \mathbf{A} & \mathbf{B}^* \mathbf{B} \end{bmatrix}^{\frac{1}{2}} = |[\mathbf{A}, \mathbf{B}]|$ be a symmetric positive semidefinite matrix. We have

$$\begin{cases} |\mathbf{A}|^2 = \mathbf{A}^* \mathbf{A} = \mathbf{E}^2 + \mathbf{G}\mathbf{G}^* \\ |\mathbf{B}|^2 = \mathbf{B}^* \mathbf{B} = \mathbf{F}^2 + \mathbf{G}^* \mathbf{G} \\ \mathbf{A}^* \mathbf{B} = \mathbf{E}\mathbf{G} + \mathbf{G}\mathbf{F}. \end{cases} \tag{13}$$

Let $\{\boldsymbol{a}_i\}_{i=1}^m$, $\{\boldsymbol{b}_i\}_{i=1}^n$ be the orthonormal eigenvectors of $|\mathbf{A}|$, $|\mathbf{B}|$, respectively. Then

$$\||\mathbf{A}|\boldsymbol{a}_i\|^2 = \langle |\mathbf{A}|^2 \boldsymbol{a}_i, \boldsymbol{a}_i \rangle = \langle (\mathbf{E}^2 + \mathbf{G}\mathbf{G}^*)\boldsymbol{a}_i, \boldsymbol{a}_i \rangle = \|\mathbf{E}\boldsymbol{a}_i\|^2 + \|\mathbf{G}^* \boldsymbol{a}_i\|^2. \tag{14}$$

Similarly,

$$\||\mathbf{B}|\boldsymbol{b}_i\|^2 = \|\mathbf{F}\boldsymbol{b}_i\|^2 + \|\mathbf{G}\boldsymbol{b}_i\|^2. \tag{15}$$

Suppose that $\|[\mathbf{A}, \mathbf{B}]\|_* = \|\mathbf{A}\|_* + \|\mathbf{B}\|_*$, then

$$
\begin{aligned}
\|\mathbf{A}\|_* + \|\mathbf{B}\|_* = \mathrm{Tr}(|\mathbf{A}|) + \mathrm{Tr}(|\mathbf{B}|) &= \sum_{i=1}^{m} \langle |\mathbf{A}|\boldsymbol{a}_i, \boldsymbol{a}_i \rangle + \sum_{i=1}^{n} \langle |\mathbf{B}|\boldsymbol{b}_i, \boldsymbol{b}_i \rangle \\
&= \sum_{i=1}^{m} \||\mathbf{A}|\boldsymbol{a}_i\| + \sum_{i=1}^{n} \||\mathbf{B}|\boldsymbol{b}_i\| \\
&= \sum_{i=1}^{m} \left( \|\mathbf{E}\boldsymbol{a}_i\|^2 + \|\mathbf{G}^*\boldsymbol{a}_i\|^2 \right)^{\frac{1}{2}} + \sum_{i=1}^{n} \left( \|\mathbf{F}\boldsymbol{b}_i\|^2 + \|\mathbf{G}\boldsymbol{b}_i\|^2 \right)^{\frac{1}{2}} \\
&\geq \sum_{i=1}^{m} \|\mathbf{E}\boldsymbol{a}_i\| + \sum_{i=1}^{n} \|\mathbf{F}\boldsymbol{b}_i\| \geq \sum_{i=1}^{m} \langle \mathbf{E}\boldsymbol{a}_i, \boldsymbol{a}_i \rangle + \sum_{i=1}^{n} \langle \mathbf{F}\boldsymbol{b}_i, \boldsymbol{b}_i \rangle \\
&= \mathrm{Tr}(\mathbf{E}) + \mathrm{Tr}(\mathbf{F}) = \mathrm{Tr}(|[\mathbf{A}, \mathbf{B}]|) = \|[\mathbf{A}, \mathbf{B}]\|_* \\
&= \|\mathbf{A}\|_* + \|\mathbf{B}\|_*
\end{aligned}
\tag{16}
$$

Therefore, both of the inequalities in this chain must be equalities, and the first one being equality only if $\mathbf{G} = \mathbf{0}$. This combined with the last equation in (13) implies

$$
\mathbf{A}^*\mathbf{B} = \mathbf{E}\mathbf{G} + \mathbf{G}\mathbf{F} = \mathbf{0} \tag{17}
$$

$\square$

## APPENDIX B   PROOF OF THEOREM 2

*Proof.* First, $l$ is defined in equation (5) as

$$
l(\mathbf{X}, \mathbf{Y}) = l([\mathbf{X}^g, \mathbf{X}^v], [\mathbf{Y}^g, \mathbf{Y}^v]) = l_g(\mathbf{Z}^g) + \lambda l_v(\mathbf{Y}^v, \hat{\mathbf{Y}}^v). \tag{18}
$$

The nonnegativity of $l_g(\mathbf{Z}^g)$ is guaranteed by Theorem 1. The validation loss $l_v(\mathbf{Y}^v, \hat{\mathbf{Y}}^v)$ is also nonnegative since it is the average (over the validation batch) of the cross entropy losses:

$$
l_v(\mathbf{Y}^v, \hat{\mathbf{Y}}^v) = \frac{1}{|B_v|} \sum_{\boldsymbol{x} \in \mathbf{X}^v} H(\delta_y, \hat{\boldsymbol{y}}) = -\frac{1}{|B_v|} \sum_{\boldsymbol{x} \in \mathbf{X}^v} \log \hat{y}_y. \tag{19}
$$

Therefore $l = l_g + \lambda l_v$ is also nonnegative.

Next, for a given $\lambda > 0$, $l(\mathbf{X}, \mathbf{Y})$ obtains its minimum value zero if and only if both $l_g(\mathbf{Z}^g)$ and $l_v(\mathbf{Y}^v, \hat{\mathbf{Y}}^v)$ are zeros.

- By Theorem 1, $l_g(\mathbf{Z}^g) = 0$ if and only if $\mathrm{span}(\mathbf{Z}_c^g) \perp \mathrm{span}(\mathbf{Z}_{c'}^g), \forall c \neq c'$.

- According to (19), $l_v(\mathbf{Y}^v, \hat{\mathbf{Y}}^v) = 0$ if and only if $\hat{\boldsymbol{y}}(\boldsymbol{x}) = \delta_y, \forall \boldsymbol{x} \in \mathbf{X}^v$, i.e., for every $\boldsymbol{x} \in \mathbf{X}_c^v$, its feature $\boldsymbol{z} = \Phi(\boldsymbol{x}; \theta)$ belongs to $\mathrm{span}(\mathbf{Z}_c^g)$.

At last, we want to prove that if $\lambda > 0$, and $\mathbf{X}^v$ contains at least one sample for each class, then $\mathrm{rank}(\mathrm{span}(\mathbf{Z}_c^g)) \geq 1$ for any $c \in \{1, \ldots, K\}$.

If not, then there exists $c \in \{1, \ldots, K\}$ such that $\mathrm{rank}(\mathrm{span}(\mathbf{Z}_c^g)) = 0$. Let $\boldsymbol{x} \in \mathbf{X}^v$ be a validation datum belonging to class $y = c$. The predicted probability of $\boldsymbol{x}$ belonging to class $c$ is defined in (3):

$$
\hat{y}_c = \mathbf{P}(\boldsymbol{x} \in c) \triangleq \left\langle \boldsymbol{z}, \frac{\mathrm{proj}_c(\boldsymbol{z})}{\max\left(\|\mathrm{proj}_c(\boldsymbol{z})\|, \varepsilon\right)} \right\rangle \bigg/ \sum_{c'=1}^{K} \left\langle \boldsymbol{z}, \frac{\mathrm{proj}_{c'}(\boldsymbol{z})}{\max\left(\|\mathrm{proj}_{c'}(\boldsymbol{z})\|, \varepsilon\right)} \right\rangle = 0. \tag{20}
$$

Thus we have

$$
l \geq \lambda l_v = -\frac{\lambda}{|B_v|} \sum_{\boldsymbol{x} \in \mathbf{X}^v} \log \hat{y}_y \geq -\frac{\lambda}{|B_v|} \log \hat{y}(\boldsymbol{x})_c = +\infty \tag{21}
$$

$\square$

## APPENDIX C    PROOF OF THEOREM 3

First, we need the following lemma.

**Lemma.** *Let $\mathbf{A} \in \mathbb{R}^{m \times n}$ be a rank-s matrix, and let $\mathbf{A} = \mathbf{U}^{(1)}\mathbf{\Sigma}^{(1)}\mathbf{V}^{(1)*}$ be the compact SVD of $\mathbf{A}$, i.e., $\mathbf{U}^{(1)} \in \mathbb{R}^{m \times s}, \mathbf{\Sigma}^{(1)} \in \mathbb{R}^{s \times s}, \mathbf{V}^{(1)} \in \mathbb{R}^{n \times s}$, then the subdifferential of the nuclear norm at $\mathbf{A}$ is:*

$$\partial \|\mathbf{A}\|_* = \left\{ \mathbf{U}^{(1)}\mathbf{V}^{(1)*} + \tilde{\mathbf{U}}^{(2)}\tilde{\mathbf{W}}\tilde{\mathbf{V}}^{(2)*} \right\}, \tag{22}$$

*where $\tilde{\mathbf{U}}^{(2)} \in \mathbb{R}^{m \times (n-s)}, \tilde{\mathbf{V}}^{(2)} \in \mathbb{R}^{n \times (n-s)}, \tilde{\mathbf{W}} \in \mathbb{R}^{(n-s) \times (n-s)}$ satisfy that the columns of $\tilde{\mathbf{U}}^{(2)}$ and $\tilde{\mathbf{V}}^{(2)}$ are orthonormal,* $\text{span}(\mathbf{U}^{(1)}) \perp \text{span}(\tilde{\mathbf{U}}^{(2)}), \text{span}(\mathbf{V}^{(1)}) \perp \text{span}(\tilde{\mathbf{V}}^{(2)})$, and $\|\tilde{\mathbf{W}}\|_2 \leq 1$.

*Proof.* Based on (6), we only need to show the following two sets are identical:

$$D_1 = \left\{ \mathbf{U}^{(1)}\mathbf{V}^{(1)*} + \mathbf{U}^{(2)}\mathbf{W}\mathbf{V}^{(2)*}, \quad \forall \mathbf{W} \in \mathbb{R}^{(m-s) \times (n-s)} \text{ with } \|\mathbf{W}\|_2 \leq 1 \right\} \tag{23}$$

$$D_2 = \left\{ \mathbf{U}^{(1)}\mathbf{V}^{(1)*} + \tilde{\mathbf{U}}^{(2)}\tilde{\mathbf{W}}\tilde{\mathbf{V}}^{(2)*}, \quad \tilde{\mathbf{U}}^{(2)}, \tilde{\mathbf{V}}^{(2)}, \tilde{\mathbf{W}} \text{ satisfy the conditions in the lemma} \right\} \tag{24}$$

On one hand, let $\boldsymbol{d} = \mathbf{U}^{(1)}\mathbf{V}^{(1)*} + \mathbf{U}^{(2)}\mathbf{W}\mathbf{V}^{(2)*} \in D_1$, and let $\mathbf{U}^{(2)}\mathbf{W} = \bar{\mathbf{U}}\bar{\mathbf{\Sigma}}\bar{\mathbf{V}}^*$ be the reduced SVD of $\mathbf{U}^{(2)}\mathbf{W} \in \mathbb{R}^{m \times (n-s)}$, i.e., $\bar{\mathbf{U}} \in \mathbb{R}^{m \times (n-s)}, \bar{\mathbf{\Sigma}} \in \mathbb{R}^{(n-s) \times (n-s)}, \bar{\mathbf{V}} \in \mathbb{R}^{(n-s) \times (n-s)}$. Then we can set $\tilde{\mathbf{U}}^{(2)} = \bar{\mathbf{U}}, \tilde{\mathbf{W}} = \bar{\mathbf{\Sigma}}\bar{\mathbf{V}}^*$, and $\tilde{\mathbf{V}}^{(2)} = \mathbf{V}^{(2)}$. It is easy to check that $\tilde{\mathbf{U}}^{(2)}, \tilde{\mathbf{V}}^{(2)}, \tilde{\mathbf{W}}$ satisfy the conditions in the lemma, and

$$\boldsymbol{d} = \mathbf{U}^{(1)}\mathbf{V}^{(1)*} + \tilde{\mathbf{U}}^{(2)}\tilde{\mathbf{W}}\tilde{\mathbf{V}}^{(2)*} \in D_2 \tag{25}$$

On the other hand, let $\boldsymbol{d} = \mathbf{U}^{(1)}\mathbf{V}^{(1)*} + \tilde{\mathbf{U}}^{(2)}\tilde{\mathbf{W}}\tilde{\mathbf{V}}^{(2)*} \in D_2$, where $\tilde{\mathbf{U}}^{(2)}, \tilde{\mathbf{V}}^{(2)}, \tilde{\mathbf{W}}$ satisfy the conditions in the lemma. Let $\tilde{\mathbf{U}}^{(2)} = \mathbf{U}^{(2)}\mathbf{P}$ and $\tilde{\mathbf{V}}^{(2)} = \mathbf{V}^{(2)}\mathbf{Q}$, where $\mathbf{P} \in \mathbb{R}^{(m-s) \times (n-s)}$ and $\mathbf{Q} \in \mathbb{R}^{(n-s) \times (n-s)}$ have orthonormal columns. After setting $\mathbf{W} = \mathbf{P}\tilde{\mathbf{W}}\mathbf{Q}^*$, we have

$$\tilde{\mathbf{U}}^{(2)}\tilde{\mathbf{W}}\tilde{\mathbf{V}}^{(2)*} = \mathbf{U}^{(2)}\mathbf{P}\tilde{\mathbf{W}}\mathbf{Q}^*\mathbf{V}^{(2)*} = \mathbf{U}^{(2)}\mathbf{W}\mathbf{V}^{(2)*}, \tag{26}$$

where $\|\mathbf{W}\|_2 \leq 1$. Therefore,

$$\boldsymbol{d} = \mathbf{U}^{(1)}\mathbf{V}^{(1)*} + \tilde{\mathbf{U}}^{(2)}\tilde{\mathbf{W}}\tilde{\mathbf{V}}^{(2)*} = \mathbf{U}^{(1)}\mathbf{V}^{(1)*} + \mathbf{U}^{(2)}\mathbf{W}\mathbf{V}^{(2)*} \in D_1 \tag{27}$$

$\square$

Now we go on to prove Theorem 3.

*Proof.* Let $\text{rank}(\mathbf{A}) = s$, and we split the computed singular vectors into two parts: $\mathbf{U}^\varepsilon = [\mathbf{U}^{(1)\varepsilon}, \mathbf{U}^{(2)\varepsilon}], \mathbf{V}^\varepsilon = [\mathbf{V}^{(1)\varepsilon}, \mathbf{V}^{(2)\varepsilon}]$, where $\mathbf{U}^{(1)\varepsilon} \in \mathbb{R}^{m \times s}, \mathbf{U}^{(2)\varepsilon} \in \mathbb{R}^{m \times (n-s)}, \mathbf{V}^{(1)\varepsilon} \in \mathbb{R}^{n \times s}$, and $\mathbf{V}^{(2)\varepsilon} \in \mathbb{R}^{n \times (n-s)}$. By the backward stability of SVD, we have $\|\mathbf{U}^{(1)} - \mathbf{U}^{(1)\varepsilon}\|_2 = O(\varepsilon/\eta)$, $\|\mathbf{V}^{(1)} - \mathbf{V}^{(1)\varepsilon}\|_2 = O(\varepsilon/\eta)$, and there exists $\tilde{\mathbf{U}}^{(2)}, \tilde{\mathbf{V}}^{(2)}$ satisfying the condition in the lemma and $\|\tilde{\mathbf{U}}^{(2)} - \mathbf{U}^{(2)\varepsilon}\|_2 = O(\varepsilon/\eta), \|\tilde{\mathbf{V}}^{(2)} - \mathbf{V}^{(2)\varepsilon}\|_2 = O(\varepsilon/\eta)$.

Because of the lemma, we have $(\mathbf{U}^{(1)}\mathbf{V}^{(1)*} + \tilde{\mathbf{U}}^{(2)}\tilde{\mathbf{V}}^{(2)*}) \in \partial\|\mathbf{A}\|_*$, and

$$
\begin{aligned}
\text{d}(\mathbf{U}^\varepsilon\mathbf{V}^{\varepsilon*}, \partial\|\mathbf{A}\|_*) \leq\ & \|\mathbf{U}^\varepsilon\mathbf{V}^{\varepsilon*} - \left(\mathbf{U}^{(1)}\mathbf{V}^{(1)*} + \tilde{\mathbf{U}}^{(2)}\tilde{\mathbf{V}}^{(2)*}\right)\|_2 \\
=\ & \|\left(\mathbf{U}^{(1)\varepsilon}\mathbf{V}^{(1)\varepsilon*} + \mathbf{U}^{(2)\varepsilon}\mathbf{V}^{(2)\varepsilon*}\right) - \left(\mathbf{U}^{(1)}\mathbf{V}^{(1)*} + \tilde{\mathbf{U}}^{(2)}\tilde{\mathbf{V}}^{(2)*}\right)\|_2 \\
\leq\ & \|\left(\mathbf{U}^{(1)\varepsilon} - \mathbf{U}^{(1)}\right)\mathbf{V}^{(1)\varepsilon*}\|_2 + \|\mathbf{U}^{(1)}\left(\mathbf{V}^{(1)\varepsilon*} - \mathbf{V}^{(1)*}\right)\|_2 \\
& + \|\left(\mathbf{U}^{(2)\varepsilon} - \tilde{\mathbf{U}}^{(2)}\right)\mathbf{V}^{(2)\varepsilon*}\|_2 + \|\tilde{\mathbf{U}}^{(2)}\left(\mathbf{V}^{(2)\varepsilon*} - \tilde{\mathbf{V}}^{(2)*}\right)\|_2 \\
=\ & O(\varepsilon/\eta)
\end{aligned}
\tag{28}
$$

$\square$

## APPENDIX D  VISUAL ILLUSTRATION OF THE SECOND TOY EXPERIMENT IN SECTION 4

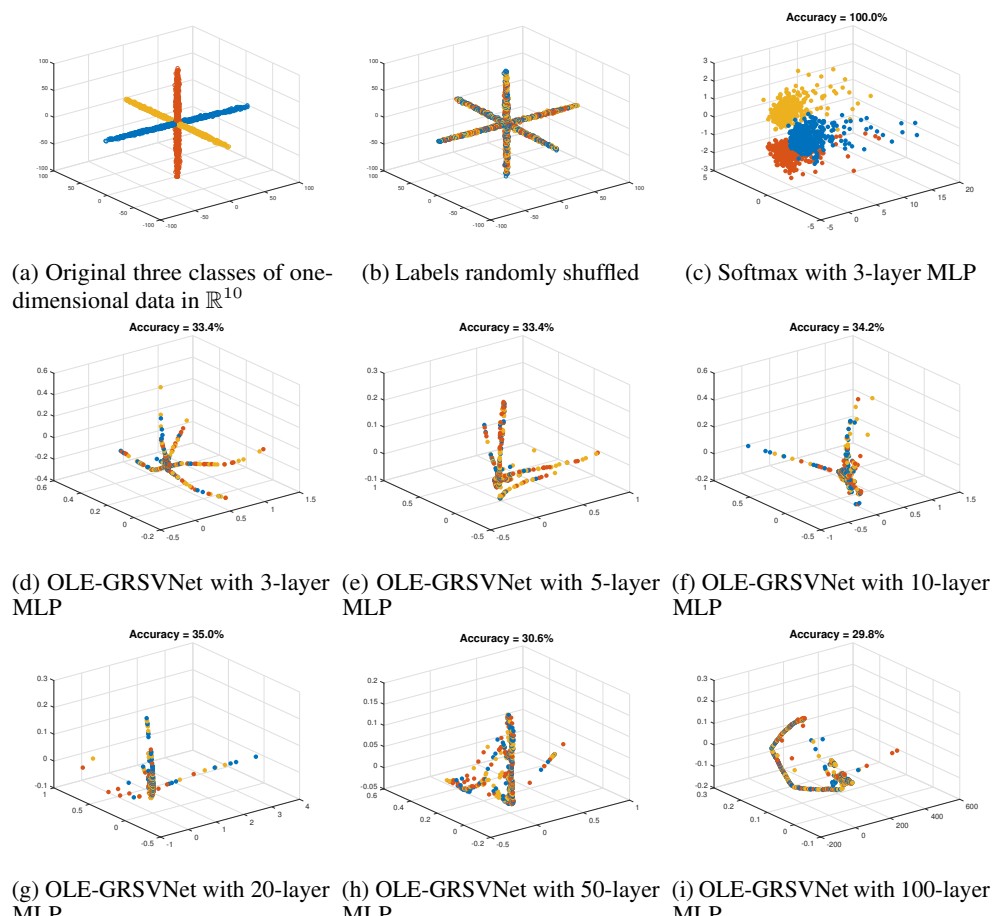

(a) Original three classes of one-dimensional data in $\mathbb{R}^{10}$

(b) Labels randomly shuffled

(c) Softmax with 3-layer MLP

(d) OLE-GRSVNet with 3-layer MLP

(e) OLE-GRSVNet with 5-layer MLP

(f) OLE-GRSVNet with 10-layer MLP

(g) OLE-GRSVNet with 20-layer MLP

(h) OLE-GRSVNet with 50-layer MLP

(i) OLE-GRSVNet with 100-layer MLP

Figure 3: Visual illustration of the second toy experiment in Section 4. (a) Three classes of one-dimensional data in $\mathbb{R}^{10}$. (b) Labels randomly shuffled. (c)-(i) Features extracted by baseline MLP with softmax classifier or OLE-GRSVNet. Only three layers of MLP are needed for conventional DNN to perfectly memorize random labels. But even with 100 layers of MLP, OLE-GRSVNet still refuses to memorize the random labels because there are no intrinsically learnable patterns.

