# OpenReview forum: "Stop memorizing: A data-dependent regularization framework for intrinsic pattern learning"
_ICLR.cc/2019/Conference_

### Official Review · AnonReviewer3 · 2018-10-25
**paper well written, but limited in novelty and significance**

**Rating:** 4
**Confidence:** 4

**Review:**

The paper proposes a data-dependent regularization method which is coupled with softmax loss to train deep neural networks for classification. The paper turns to Orthogonal Low-rank Embedding (OLE) loss for the geometric constraint that one class of data/feature are assumed to reside on a low-rank subspace that subspaces of different classes are orthogonal ideally. The probability in the softmax is then modeled as cosine similarity between data feature and the class-specific subspaces. In this way, geometric loss and softmax loss have the common goal for optimization. Moreover, during training, the geometry enforced on one batch of features is simultaneously validated on a separate batch using a validation loss. The experiments seem to suggest such a model helps avoid overfitting/memorizing noisy training data. The paper reads well and is easy to follow.

However, the paper is limited in technical novelty and practical significance. Here are some concerns --

1) The paper only studies one method based on OLE, though it cites the center loss [19]. How does the center loss behave in face of noisy training label? Would it also be able to refuse to fit the noisy training data?

2) As each class has its own (low-rank) subspace, and the rank is reduced by imposing the nuclear norm. It seems that the proposed method is hard to extend to many classes (class number is larger than the dimension)?

3) The datasets in the experiments are quite small in scale and class number. It is not persuasive unless tested on larger scale data or with large class number.

4) The proposed method seems to be limited in deal with discrete labels (e.g., classification), is it easy to extend to continuous target, say regression problems like depth estimation and surface normal estimation?

5) While the authors claim as a main contribution that the proposed GRSVNet is a general framework, it is hard to see how this framework can be used in other tasks other than classification.

6) The experiments are less persuasive. It's better to add the error bar to see the improvement by the proposed method is not due to random initialization. Running time should also be compared, as nuclear norm seems to be time consuming.

---

### Official Review · AnonReviewer2 · 2018-11-02
**Model has high bias and low variance**

**Rating:** 4
**Confidence:** 3

**Review:**

The paper proposes a framework for data-dependent DNN regularization which claimed to be capable of producing highly discriminative features residing in orthogonal-low-rank subspaces. The main claim is that the proposed regularization makes the neural network not memorizing from the training data and motivate learning the intrinsic patterns.  The experiments were done with three image dataset.

The main problem with this paper is the low training accuracy, but the high testing accuracy (Table 1). This implies that the model has a high bias and low variance. Intuitively the model is consistently predicting a wrong target function (probably from the self-validation).

---

### Official Review · AnonReviewer1 · 2018-11-05
**Stop memorizing: A data-dependent regularization framework for intrinsic pattern learning**

**Rating:** 7
**Confidence:** 4

**Review:**

Previous works have shown that DNNs are able to memorize random training data, even ignoring the enforcing of data-dependent geometric regularization constraints. In this work, authors show convincing results indicating that this is due to a lack of consistency between the main classification loss (typically soft-max cross entropy) and the selected geometric constraint. Consequently, they propose a simple approach where the softmax loss is replaced by a validation loss that is consistent with the enforced geometry. Specifically, for each training batch, instead of considering a join loss (soft-max cross entropy + geometric constraint), they apply a sequential process, where each training batch is split into two sub-batches: a first batch used to apply the geometric constraint, and a second batch (based on the proposed feature geometry) where a validation loss is used to generate a predicted label distribution. Authors test the proposed idea using an implementation that enforces that samples from each class belong to an independent low-rank sub-space (enforced geometric constraint). Results verify the main hypothesis. Specifically, the resulting model is able to fit real data but not data with random labels. The strength of this evaluation is enhanced including results from relevant baselines. In terms of generalization of real data, the proposed approach offers a small increase in accuracy.

Paper is well written, main hypothesis is relevant, and results are convincing. While not a complete answer to the main questions related to the abilities of DNNs to fit and generalize on real data, this paper offers relevant insights related to the role of finding/using a suitable loss function to train DNNs. These results are relevant to the community and they can illuminate future work, so this reviewer recommend to accept this paper.

---

### Public Comment · (anonymous) · 2018-11-23
**Another recent and similar data-dependant regularization method**

Another concurrent ICLR submission  also seems to have a very similar idea of low rank embeddings and its advantages. Could you also possibly mention relation with that work and see if the effects observed in that paper are also observed here ?

[1] Anonymous, https://openreview.net/forum?id=SJzvDjAcK7.

---

> ### Comment · Area_Chair1 · 2018-11-29
> **link broken**
>
> Thanks for the comments! The link you provided is missing. Could you give another link? Thanks!

---

> > ### Comment · AnonReviewer3 · 2018-11-30
> > **https://openreview.net/forum?id=SJzvDjAcK7**
> >
> > It works by removing the period.

---

> > > ### Public Comment · (anonymous) · 2018-12-03
> > > **Thanks**
> > >
> > > Ah yes. Sorry about that. It does work by removing the period.

---

### Meta-Review · Area_Chair1 · 2018-12-08

**Confidence:** 4
**Recommendation:** Reject

**Metareview:**

The paper proposes an interesting data-dependent regularization method for orthogonal-low-rank embedding (OLE). Despite the novelty of the method, the reviewers and AC note that it's unclear whether the approach can extend other settings with multi-class or continuous labels or other loss functions.